

# Evaluation of high-resolution predictions of fine particulate matter and its composition in an urban area using PMCAMx-v2.0

Brian T. Dinkelacker[1], Pablo Garcia Rivera[1], Ioannis Kioutsioukis[2], Peter J. Adams[3,4], Spyros N. Pandis[5,6]

[1]*Department of Chemical Engineering, Carnegie Mellon University, Pittsburgh, PA, 15213*
[2]*Department of Physics, University of Patras, 26500, Patras, Greece*
[3]*Department of Civil and Environmental Engineering, Carnegie Mellon University, Pittsburgh, PA, 15213*
[4]*Department of Engineering and Public Policy, Carnegie Mellon University, Pittsburgh, PA, 15213*
[5]*Institute of Chemical Engineering Sciences (FORTH/ICE-HT), 26504, Patras, Greece*
[6]*Department of Chemical Engineering, University of Patras, 26500, Patras, Greece*

*Correspondence to*: Spyros N. Pandis (spyros@chemeng.upatras.gr)

**Abstract**

Accurately predicting urban $PM_{2.5}$ concentrations and composition has proved challenging in the past, partially due to the resolution limitations of computationally intensive chemical transport models (CTMs). Increasing the resolution of $PM_{2.5}$ predictions is desired to support emissions control policy development and address issues related to environmental justice. A nested grid approach using the CTM PMCAMx-v2.0 was used to predict $PM_{2.5}$ at increasing resolutions of 36 x 36, 12 x 12, 4 x 4, and 1 x 1 km for a domain largely consisting of Allegheny County and the city of Pittsburgh in southwestern Pennsylvania, US during February and July 2017. Performance of the model in reproducing $PM_{2.5}$ concentrations and composition was evaluated at the finest scale using measurements from regulatory sites as well as a network of low-cost monitors. Total $PM_{2.5}$ mass is reproduced well by the model during the winter period with low fractional error (0.3) and fractional bias (+0.05) when compared to regulatory measurements. Comparison with speciated measurements during this period identified small underpredictions of $PM_{2.5}$ sulfate, elemental carbon (EC), and organic aerosol (OA) offset by a larger overprediction of $PM_{2.5}$ nitrate (bias = +1.4 µg m$^{-3}$, fractional bias = +0.81). In the summer period, total $PM_{2.5}$ mass is underpredicted with fractional bias of -0.39. Here, $PM_{2.5}$ nitrate is overpredicted again with a large fractional bias (+0.7) but significantly lower magnitude (+0.4 µg m$^{-3}$).





Underpredictions in PM$_{2.5}$ sulfate and EC contribute to the negative prediction bias of total
PM$_{2.5}$ (-0.4 µg m$^{-3}$ and -0.2 µg m$^{-3}$, respectively), however the largest underprediction is
seen for summer OA (bias = -1.9 µg m$^{-3}$, fractional bias = -0.41). In the winter period, the
model performs well reproducing the variability between urban measurements and rural
measurements of local pollutants such as EC and OA. This effect is also captured well in
the summer for EC, although the OA performance here is less consistent because much
more of this OA is secondary and transported from outside of the inner modeling domain.
Comparison with total PM$_{2.5}$ concentration measurements from low-cost sensors yielded
similar results with slightly higher overpredictions seen in the winter (fractional bias =
+0.24) and lower underpredictions seen in the summer (fractional bias = -0.27).
Inconsistencies in PM$_{2.5}$ nitrate predictions in both periods are believed to be due to errors
in partitioning between PM$_{2.5}$ and PM$_{10}$ modes and motivate improvements to the treatment
of dust particles within the model. The underprediction of summer OA would likely be
improved by updates to biogenic SOA chemistry within the model, which would result in
an increase of long-range transport SOA seen in the inner modeling domain. These
improvements are obvious topics for future work towards model improvement.
Comparison with regulatory monitors showed that increasing resolution from 36 km to 1
km improved both fractional error and fractional bias by 0.04 in February 2017.  In July
2017, fractional error decreased by 0.05 and fractional bias improved by 0.07 with
increasing resolution. Improvements at all types of measurement locations indicated an
improved ability of the model to reproduce urban-rural PM$_{2.5}$ gradients at higher
resolutions.

**1 Introduction**

Fine particulate matter with aerodynamic diameter less than 2.5 µm (PM$_{2.5}$) has
been associated with public health concerns due to short and long-term exposure. Some of
the health effects of PM$_{2.5}$ include increased risk of heart disease, increased likelihood of
heart attacks and strokes, impaired lung development, and increased risk of lung disease
(Dockery and Pope, 1994). Chemical transport models are frequently used for supporting
the development of air quality policies designed to protect public health. To evaluate these





policies, CTMs must simulate $PM_{2.5}$ concentrations and their response to changes in
emissions accurately.
Grid resolution is an important factor for CTM studies focusing on major urban
areas since on-road traffic, commercial cooking, and biomass burning can have sharp
gradients at the urban scale (Lanz et al., 2007; Allan et al., 2010). High spatial resolution
measurements of $PM_1$ in the city of Pittsburgh in high source-impact locations are on
average 40% higher than at urban background locations (Gu et al., 2018). Heightened
organic aerosol concentrations have been observed in commercial districts containing
multiple restaurants (Robinson et al., 2018). The demographic characteristics of the
population can also have large variations at the neighborhood scale. High resolution
predictions of pollutant concentrations allow for exposure assessments that compare
subpopulations within the same metropolitan area to answer environmental justice related
questions (Anand, 2002). The benefits of high-resolution modeling must be balanced with
the increased complexity in the development of accurate, high-resolution emission
inventories and increased computational cost and storage requirements.
Previous studies have found small to modest improvements on the predictive ability
of regional CTMs for ozone in the summers of 1995, 1996, and 1997 moving from 36 km
to 12 km resolution (Arunachalam et al., 2006) as well as in July 1988 using a dynamic
grid system with sizes varying from 18.5 km to 4.625 km (Kumar and Russell, 1996).
Stroud et al. (2011) found that the accurate simulation of urban and large industrial plumes
required a grid resolution of 2.5 km in order to properly capture contributions from local
sources of primary organic aerosol (POA) and volatile organic compounds (VOCs).
Zakoura and Pandis (2019) investigated the effect of increasing grid resolution on $PM_{2.5}$
nitrate predictions and found that increasing the resolution to 4 km reduced bias by 65%.
Fountoukis et al. (2013) reported a reduction of the bias for black carbon (BC)
concentrations in the northeastern US when the grid resolution was reduced from 36x36
km to 4x4 km. Pan et al., (2017) allocated county-based emissions at 4 km and 1 km grid
resolution using the default approach from the National Emissions Inventory and found
small changes in model performance for $NO_x$ and ozone. The 1 km simulation was able to
resolve the detailed spatial variability of emissions in heavily polluted areas including
highways, airports and industrially focused sub-regions.





One of the weaknesses of several of the above studies has been that the gridded
emissions used at the higher resolutions were the results of interpolation. It is not clear if
the remaining discrepancies between model predictions and measurements were due to
errors in the spatial distribution of the high-resolution emissions, errors in the overall
magnitude of the emissions over an urban area or other modeling errors in the simulation
of various processes (chemistry, condensation/evaporation, etc.). It is also not clear if errors
in previous simulations of urban $PM_{2.5}$ are due to inaccuracies in the transport of regional
$PM_{2.5}$ to urban areas. In this work, we explore the impacts of increasing the resolution of
emissions inputs and CTM output on $PM_{2.5}$ predictions in southwestern Pennsylvania
during the months of February and July 2017, including the ability of the model to
reproduce observed differences between urban and rural $PM_{2.5}$ at the various grid
resolutions.
Garcia Rivera et al. (2022) investigated the effects of increasing grid resolution of
model inputs and CTM output on source resolved predictions of $PM_{2.5}$ concentration and
population exposure at 36 km, 12 km, 4 km, and 1 km. Moving to 12 x 12 km resolution
resolved much of the urban-rural gradient. Increasing to 4 x 4 km resolved stationary
sources such as power plants and the 1 x 1 km resolution results revealed intra-urban
variations and individual roadways. Regional pollutants with low spatial variability such
as $PM_{2.5}$ nitrate showed modest changes when increasing the resolution to 4 x 4 km and
higher. Local pollutants such as black carbon and organic aerosol showed gradients that
were only resolved at the finest resolution. The ability of these simulations to reproduce
$PM_{2.5}$ concentrations at different resolutions is evaluated here against multiple
measurement sources and types.
We apply the Particulate Matter Comprehensive Air quality Model with Extensions
version 2.0 (PMCAMx-v2.0) to study the impact of increasing model resolution on the
ability to reproduce observed $PM_{2.5}$ concentrations. We evaluate the PMCAMx predictions
at various grid resolutions against regulatory measurements of $PM_{2.5}$ concentration and
composition, as well as measurements from a network of low-cost sensors (Zimmerman et
al., 2018) during February and July 2017 which provide a unique opportunity for
comparison not available to previous studies. Aerosol mass spectrometer (AMS)





measurements taken in Pittsburgh during February 2017 were also used to evaluate model
predictions.

**2 Model Description**

PMCAMx-v2.0, the Particulate Matter Comprehensive Air Quality Model with
Extensions (Karydis et al., 2010; Murphy and Pandis, 2010; Tsimpidi et al., 2010) is a
state-of-the-art atmospheric chemical transport model (CTM) that uses the framework of
the CAMx model (Environ, 2006) with advanced aerosol chemistry modules. To track the
dynamic evolution of aerosol mass, 10 moving size sections are used (Gaydos et al., 2003).
The chemical mechanism SAPRC99 (Carter, 1999) was used for gas-phase chemistry,
including 237 individual chemical reactions involving 91 chemical species. Aqueous-phase
chemistry is calculated with the Variable Size Resolution Model (Fahey and Pandis, 2001).
PMCAMx-v2.0 considers the formation of aerosol mass comprised of sulfate, nitrate,
ammonium, sodium, chloride, water, elemental carbon, as well as lumped organic species
(both primary and secondary). Inorganic aerosol growth is modelled using an approach that
assumes equilibrium between the bulk aerosol and gas phases. Partitioning of semivolatile
inorganic aerosol is calculated using ISORROPIA-I (Nenes et al., 1998). The Volatility
Basis Set (VBS) was used to calculate partitioning of organic aerosol components across a
distribution of species volatility (Donahue et al., 2006). Volatility bins (10) with effective
saturation concentration from $10^{-3}$ to $10^{6}$ µg m$^{-3}$ (at 298 K) are used for primary organic
aerosol (POA). Secondary organic aerosol is split into anthropogenic (aSOA) and biogenic
(bSOA) components, formed from a variety of SOA-forming volatile organic compounds
(VOCs) from human activity and natural sources, respectively using $NO_x$-dependent SOA
formation yields (Lane et al., 2008). Both aSOA and bSOA are split into 4 volatility bins
with effective saturation concentration from $10^{0}$ to $10^{3}$ µg m$^{-3}$ (at 298 K).

**3 Model Application**

Air quality simulations of a 5184 km$^2$ area comprised of southwestern Pennsylvania
and smaller parts of eastern Ohio and norther West Virginia were performed using
PMCAMx. Two distinct simulation periods of February and July 2017 were investigated.
The approach of Garcia et al. (2022) was used to produce speciated PM$_{2.5}$ concentration




predictions at spatial resolution of 36 km, 12 km, 4 km, and 1 km. Surface-level boundary
conditions for the 36 x 36 km simulations are provided in Table S1. Boundary conditions
for the higher resolution grids are taken from the results of parent-grid simulations. The
first two days of simulation output have been removed from the analysis to allow for model
spin-up.

Meteorological fields were calculated using the Weather Research and Forecasting

model (WRF-v3.6.1) with horizontal resolution of 12 x 12 km, providing wind
components, eddy diffusivity, temperature, pressure, humidity, clouds, and precipitation
inputs for use in PMCAMx. Meteorology initial and boundary conditions were retrieved
from the ERA-Interim global climate re-analysis database. The United States Geological
Survey database was used to obtain input data for terrain, land-use, and soil type. When
necessary, WRF output was interpolated to higher resolutions. An evaluation of
interpolated meteorological inputs using data from METAR stations near the city of
Pittsburgh in southwestern Pennsylvania determined that errors in the magnitude and
phasing of diurnal cycles of temperature, relative humidity, and wind speed are
appropriately small for use in air quality studies. These results are provided in the
supplementary material (Fig. S1, S2).

Anthropogenic emissions are derived from the 2017 projections of the 2011

National Emissions Inventory (Eyth and Vukovich, 2016) modelling platform. The Sparse
Matrix Operator Kernel Emissions modeling system (SMOKE) was used, along with
meteorological inputs to calculate emissions at a horizontal resolution of 12 x 12 km.
Default spatial surrogates were used to allocate these emissions to higher resolutions.
Custom surrogates were developed for commercial cooking and on-road traffic emissions
sectors and used for the primary analysis in this work.

For commercial cooking, the normalized restaurant count was used to distribute the

emissions from the sector in space within the 1 x 1 km and 4 x 4 km domains. This surrogate
distributed commercial cooking emissions based on the density of restaurants identified by
the Google Places Application Programming Interface. To allocate on-road traffic
emissions, the output from the traffic model of Ma et al. (2020) was used. This model
simulates hourly traffic using data from the Pennsylvania Department of Transportation.
Emissions from the on-road traffic sector were then allocated based on these values.



### 3.1 Available measurements for model evaluation

Model predictions of sulfate, nitrate, elemental carbon and organic aerosol were compared with measurements from 4 sites from the EPA Chemical Speciation Network (EPA-CSN) (U.S. EPA, 2002). The locations of these 4 sites are shown in Figure 1a. These sites include: Lawrenceville, an urban background site 4 km northeast of downtown Pittsburgh; Hillman State Park located in a state park in southwest Pennsylvania in a rural and remote location approximately 40 km upwind of Pittsburgh; Steubenville in the Ohio River Valley close to industrial installations and coal-fired power plants, and the Liberty-Clairton monitor, which is located close to the Clairton Coke Works in the Monongahela River Valley 14 km southeast of downtown Pittsburgh. Speciated $PM_{2.5}$ measurements from EPA-CSN sites are available every three days during the simulation periods. Daily non-speciated measurements of total $PM_{2.5}$ mass concentration are available from 17 sites within the inner simulation domain and are used to further evaluate total $PM_{2.5}$ mass concentration predictions. The locations of these sites are also shown in Figure 1a.

For February 2017, high-resolution AMS measurements from the Carnegie Mellon University supersite (Gu et al., 2018) are used to evaluate the predicted chemical composition of $PM_{2.5}$ model predictions. Positive matrix factorization results are also used to investigate the breakdown of organic aerosol components. AMS measurements were taken continuously from February 1 to February 14, 2017. Due to uncertainties with the AMS collection efficiency during this campaign, we use here only the fractional particle composition data.

PMCAMx predictions of $PM_{2.5}$ were also compared with measurements taken with a network of Real-time Affordable Multi-Pollutant (RAMP) monitors (Zimmerman et al., 2018) distributed in the city of Pittsburgh. During the winter period measurements at 7 sites were available, all located within the boundaries of the city of Pittsburgh, while 22 sites were in operation during the summer period with a few sites also outside the city (Fig. 1b). Uncertainty in these low-cost measurements of $PM_{2.5}$ mass concentration is between 3-4 $\mu g\ m^{-3}$ for hourly averaging times (Malings et al., 2019).

The model performance is assessed in terms of the mean bias (BIAS), the mean error (ERROR), the fractional bias (FBIAS) and the fractional error (ERROR):


$\quad$ $\text{BIAS} = \frac{1}{N}\sum_{i=1}^{N} P_i - O_i$ $\hspace{4cm}$ (1)
$\quad$ $\text{FBIAS} = \frac{2}{N}\sum_{i=1}^{N}\frac{P_i - O_i}{P_i + O_i}$ $\hspace{4cm}$ (2)
$\quad$ $\text{ERROR} = \frac{1}{N}\sum_{i=1}^{N}|P_i - O_i|$ $\hspace{4cm}$ (3)
$\quad$ $\text{FERROR} = \frac{2}{N}\sum_{i=1}^{N}\frac{|P_i - O_i|}{P_i + O_i}$ $\hspace{4cm}$ (4)

$\quad$ where $N$ is the number of valid measurements, $P_i$ is the predicted concentration and $O_i$ is
$\quad$ the corresponding observed concentration. The fractional error metric is bounded by 0
$\quad$ (perfect prediction performance) and 2.0 (extremely poor prediction performance).
$\quad$ Fractional bias is bounded by -2.0 (extreme underprediction) and +2.0 (extreme
$\quad$ overprediction).

$\quad$ **4 Evaluation of high-resolution model performance**
$\quad$ **4.1 Winter**
$\quad$ $\quad$ Table 1 summarizes the performance metrics of daily average PMCAMx-v2.0
$\quad$ PM$_{2.5}$ predictions in the 1x1 km resolution, when compared with daily measurements from
$\quad$ EPA regulatory PM$_{2.5}$ monitors. The speciated performance is illustrated in Figure 2.
$\quad$ Predictions of total PM$_{2.5}$ mass perform well against regulatory measurements in the
$\quad$ February simulation period, with fractional error of 0.3 and fractional bias of +0.07.
$\quad$ $\quad$ Average measured PM$_{2.5}$ sulfate for this time period was 1.9 µg m$^{-3}$. Lower sulfate
$\quad$ levels were observed at the Lawrenceville site in Pittsburgh (1.2 µg m$^{-3}$) while significantly
$\quad$ higher levels were observed at the Steubenville site (3.1 µg m$^{-3}$). Predicted domain-average
$\quad$ PM$_{2.5}$ sulfate at 1 x 1 km resolution was 1.3 µg m$^{-3}$. Overall fractional error for sulfate
$\quad$ predictions was 0.41 and no overall bias was observed (fractional bias of -0.02). PM$_{2.5}$
$\quad$ sulfate was slightly overpredicted at Hillman State Park (+0.18 fractional bias) and
$\quad$ Lawrenceville (+0.25 fractional bias) and underpredicted at the industrial sites,
$\quad$ Steubenville (-0.24 fractional bias) and Liberty/Clairton (-0.43 fractional bias) where
$\quad$ observed PM$_{2.5}$ sulfate concentrations were higher.



Overpredictions were seen for $PM_{2.5}$ nitrate, with a fractional bias of +0.81. The
average measured concentration at EPA-CSN sites within the simulation domain was 1.5
µg m$^{-3}$, while the domain-average predicted concentration was 1.8 µg m$^{-3}$. Observed
average $PM_{2.5}$ nitrate concentrations at Hillman State Park and Lawrenceville were slightly
lower at 1.1 µg m$^{-3}$ and 1.2 µg m$^{-3}$, respectively. Nitrate at the Steubenville location was
observed to be higher on average at 2.2 µg m$^{-3}$. This overprediction is seen at all sites but
is particularly prevalent at Hillman State Park, Lawrenceville, and Liberty/Clairton, where
errors are of the order of a factor of two. Previous PMCAMx modeling studies have found
similar over-predictions. Part of this overprediction was due to the use of coarse-grid
resolution (Zakoura and Pandis, 2018), but this is unlikely to be the cause here, because
81% of the predicted domain-average nitrate is transported from outside of the inner
modeling domain. These inconsistencies in $PM_{2.5}$ nitrate predictions are likely due to errors
in the partitioning of nitrate between the fine ($PM_{2.5}$) and coarse ($PM_{10}$) modes, resulting
in an overprediction of $PM_{2.5}$ nitrate. Resolving this modeling error likely requires
improvements to the treatment of dust within the model, and the use of a dynamic approach
for inorganic aerosol calculations rather than the bulk equilibrium approach.
The behavior of $PM_{2.5}$ ammonium measurements is similar to that of nitrate as most
of it is in the form of ammonium nitrate. The average measured concentration at the four
EPA-CSN stations was 0.9 µg m$^{-3}$. At Hillman State Park and Lawrenceville, the measured
average was lower at 0.5 µg m$^{-3}$ but higher at the Liberty/Clairton location at 2.1 µg m$^{-3}$.
$PM_{2.5}$ ammonium was overpredicted similarly to $PM_{2.5}$ nitrate with +0.83 fractional bias.
The average measured concentration of $PM_{2.5}$ elemental carbon at EPA-CSN sites during
February 2017 was 1.1 µg m$^{-3}$. Elemental carbon concentrations are more localized than
the inorganic $PM_{2.5}$ components. At Hillman State Park the average measured
concentration was only 0.5 µg m$^{-3}$ while at Liberty/Clairton the averaged measured
concentration was 2.9 µg m$^{-3}$. For elemental carbon, the predicted domain-average was 0.4
µg m$^{-3}$. Average elemental carbon concentration in the 4 x 4 km simulation grid outside of
the inner modeling domain was 0.3 µg m$^{-3}$. Black carbon predictions at all sites had a
fractional error of 0.71 with fractional bias of -0.08. Elemental carbon was overpredicted
at the urban site with fractional bias of 0.73 and underpredicted at the other sites.





Average measured OA during this period was 4.4 µg m$^{-3}$, but with significant
spatial variability. At Hillman State Park and Lawrenceville measured OA was 3.1 µg m$^{-3}$
and 3.4 µg m$^{-3}$, respectively. At Liberty/Clairton and Steubenville the average measured
OA was 7 µg m$^{-3}$ and 6.3 µg m$^{-3}$, respectively. Domain-average predicted OA was 2.2 µg
m$^{-3}$. Outside of the inner 1 x 1 km domain, average predicted OA was 1.6 µg m$^{-3}$,
suggesting that the majority of predicted OA is transported from outside of the 1 x 1 km
grid. Overall OA prediction performance in the winter is acceptable at 0.53 fractional error
and low fractional bias (-0.01). At individual sites, performance varies. OA is predicted
with low fractional bias (-0.10) at the rural Hillman State Park site. OA is overpredicted by
with +0.31 fractional bias at the urban site in Lawrenceville and underpredicted at both
industrial sites. An added degree of uncertainty exists with the industrial sites within the
inner domain. The emissions from these sources may be underestimated in the inventory
and these locations are also difficult to accurately model due to their geographic location
in river valleys.
Average concentrations of PM$_{2.5}$ sulfate, nitrate, and ammonium in the 4 x 4 km
resolution domain were around 83% of the average predicted concentrations in the inner 1
x 1 km simulation grid. For elemental carbon and OA, the outer concentration was 64%
and 73% of the inner concentration respectively, indicating that these species had
significant local sources. For these more local pollutants, the model appears to perform
well in terms of capturing urban-rural gradients, but with a tendency towards
underprediction at the rural site in Hillman State Park and overprediction at the urban site
in Lawrenceville. The model also underpredicts EC and OA at the industrial locations,
especially elemental carbon (-0.67 and -1.02 fractional bias at Steubenville and
Liberty/Clairton, respectively). This again suggests errors in the emissions inventory or
problems in simulating atmospheric dispersion near the sources.
Comparisons with the PM$_1$ composition as determined by the AMS from February
3 through February 14, 2017, show excellent agreement for all species (Fig. 3a). Gu et al.
(2018) used PMF analysis and allocated total measured OA into five factors. Three of them
corresponded to primary organic aerosol: hydrocarbon-like OA (HOA), cooking OA
(COA) and biomass burning OA (BBOA) and two secondary OA factors: more-oxidized
organic aerosol (MO-OOA) and less-oxidized organic aerosol (LO-OOA). To compare





PMCAMx predictions with the primary PMF factors, two additional simulations were
performed in which emissions from biomass burning and commercial cooking were set to
zero. The predicted concentrations were then subtracted from the base case to estimate the
contribution from each respective source. The remaining primary OA was assigned to
HOA. The LO-OOA and MO-OOA factors were added together and compared with the
PMCAMx SOA predictions.

The predicted cooking OA (COA) at the CMU site is 25% of the total OA and is in
agreement with the PMF/AMS estimate of 22% (Fig. 3b). This is encouraging given the
small bias of the model for total OA levels. The predicted HOA and BBOA are higher than
measured by a factor of 2 or more. At the same time, the measurements indicate a
surprisingly high contribution of SOA (53% of the total OA) during a period with little
photochemical activity and low levels of OH radicals. SOA is predicted to be just 20% of
the total during this time period. These discrepancies may indicate transformation of the
HOA and BBOA to OOA during this wintertime period, that are not included in the model.
Kodros et al. (2021) recently suggested that BBOA can react with the $NO_3$ during the
winter and can be transformed to OOA.

**4.2 Summer**

Total $PM_{2.5}$ mass concentrations are underpredicted in the summer period. The
average measured $PM_{2.5}$ value in the regulatory network in the area was 11.4 µg m$^{-3}$, while
the average predicted value at the regulatory sites was 4 µg m$^{-3}$ lower.

Speciated $PM_{2.5}$ performance is illustrated in Figure 4. Average measured $PM_{2.5}$
sulfate for the summer period was 2 µg m$^{-3}$. Slightly lower levels were observed at the
Lawrenceville site in Pittsburgh (1.9 µg m$^{-3}$). Liberty/Clairton had higher measured sulfate
concentrations (2.6 µg m$^{-3}$), but this difference between locations is lower than what was
observed in the winter period. Predicted domain-average $PM_{2.5}$ sulfate at 1 x 1 km
resolution was 1.3 µg m$^{-3}$. Overall fractional error (0.62) and fractional bias (-0.21) for
sulfate predictions was higher than in the winter simulation period. $PM_{2.5}$ sulfate was
underpredicted at all sites but to the largest extent at Hillman State Park (-0.36 fractional
error).



Overpredictions of $PM_{2.5}$ nitrate were also seen the summer period, and at all types

of sites. Average measured $PM_{2.5}$ nitrate was 0.3 µg m$^{-3}$, much lower than in the winter.
The domain-average predicted $PM_{2.5}$ nitrate was 0.7 µg m$^{-3}$. Again, predicted $PM_{2.5}$ nitrate
in the inner domain is dominated by material transported from outside the boundaries
(75%), so the issue is not resolved by using a high-resolution grid. Improvements to $PM_{2.5}$
nitrate formation are needed in the form of dust models with increased complexity to
resolve the issues with fine-coarse mode partitioning of particulate nitrate. These issues
have been highlighted by decreased concentrations of $PM_{2.5}$ pollution in recent years.

Observed $PM_{2.5}$ ammonium concentrations at EPA-CSN sites were also much

lower in the summer with an average value of 0.5 µg m$^{-3}$. Slightly higher average
concentrations were observed at Liberty/Clairton (0.7 µg m$^{-3}$) and slightly lower
concentrations were observed at Steubenville (0.4 µg m$^{-3}$). The domain-average predicted
$PM_{2.5}$ ammonium concentration was 0.6 µg m$^{-3}$. The average concentration directly outside
of the inner domain was 0.5 µg m$^{-3}$. Overall performance was better for ammonium in the
summer than in the winter with fractional error of 0.62 and fractional bias of +0.44. The
strongest overprediction is seen at the Steubenville site (+0.57 fractional bias).

The average measured elemental carbon (EC) concentration in July was 0.7 µg m$^{-3}$

Measured EC carbon was significantly higher at Liberty/Clairton (1 µg m$^{-3}$) and lower
at rural Hillman State Park (0.4 µg m$^{-3}$). Domain-average predicted EC was 0.3 µg m$^{-3}$.
Outside of the inner domain, the average predicted concentration was 0.2 µg m$^{-3}$. Elemental
carbon predictions in July had a lower fractional error compared to the winter at 0.60 but
showed a stronger negative fractional bias at -0.33. The model severely underpredicts at
Hillman State Park (-0.86 fractional bias), where measured concentrations were lowest, but
also at the industrial sites of Steubenville (-0.55 fractional bias) and Liberty/Clairton (-0.65
fractional bias). EC was slightly overpredicted at the urban Lawrenceville location (+0.14
fractional bias). While the urban-rural gradient in EC is slightly overpredicted, the model
is still able to capture well the variability between rural (Hillman State Park) and urban
(Lawrenceville). The model struggles to reproduce high measurements of EC at the
Steubenville site, reiterating the issues with industrial EC seen in the winter.

Average measured OA concentration was 4.5 µg m$^{-3}$ in July. Higher concentrations

were observed at the industrial sites, Liberty/Clairton and Steubenville (5.0 µg m$^{-3}$)





respectively. The lowest observed concentration was in Hillman State Park (3.6 µg m$^{-3}$).
The average predicted concentration at CSN sites was 2.7 µg m$^{-3}$. On average, OA is
underpredicted with fractional bias of -0.47. This underprediction occurs at all sites but is
less prevalent at the urban Lawrenceville location (-0.19 fractional bias) and is most
dramatic in Steubenville (-0.65 fractional bias). Because such a large fraction of the OA in
the summer is predicted to be secondary (50% of local OA on average) and transported
from outside of the inner modeling domain (84% of total OA), treatment of SOA formation
is likely a key factor contributing to the underprediction of PM$_{2.5}$ in the summer. While
these improvements are necessary for overall model improvement, they do not have
significant impact on the urban-rural gradients which are the focus of this work and are
driven by primary species. The performance of EC predictions in various locations is
encouraging with regards to primary PM$_{2.5}$ performance.

**5 Effect of grid resolution on PM$_{2.5}$ performance**
To determine the effect of grid resolution on the ability of the model to resolve
geographical variations in PM$_{2.5}$ concentrations, daily average measurements from the 17
EPA regulatory sites were compared with PMCAMx predictions from simulations at 36
km, 12 km, 4 km and 1 km. The PMCAMx performance metrics are summarized in Table

2.


**5.1 Winter**
During the winter period, increasing grid resolution reduces the average fractional
error from 34% at 36 x 36 km to 30% at 1 x 1 km. The higher resolution also improved the
fractional bias, from -0.09 at 36 x 36 km to +0.05 at 1 x 1 km. The performance is illustrated
in Figure 5. Performance at urban locations stayed steady in the winter, with fractional
error changing from 0.30 to 0.26 and fractional bias changing from +0.02 to +0.08 moving
from 36 km to 1 km resolution (Fig. S3). Rural performance improved to a greater extent,
with fractional error improving from 0.33 to 0.28 and fractional bias lowering from +0.21
to +0.11.
The comparison with low-cost sensor measurements largely represents the
performance of the model in terms of urban PM$_{2.5}$ predictions. The performance metrics of



PMCAMx-v2.0 when compared to measurements from low-cost sensors are shown in
Table 3. Moving from low to high resolution, the predictions go from no bias (-0.02) to a
bias of +0.24. Due to the slight overprediction of the urban-rural gradient seen earlier
(particularly with EC), the high resolution would likely lead to more positive biases when
compared to a largely urban network. Fractional error increases slightly, but still exhibits
good performance moving from 0.33 to 0.37.

**5.2 Summer**

In the summer period, (Fig. 6) the model performance improved as the resolution
increased from 36 km to 1 km. Fractional error decreased from 0.53 to 0.48, while
fractional bias increased from -0.46 to -0.39. In July, performance at the urban locations
significantly increased with resolution (Fig. S4). Fractional error decreased from 52% at
36 x 36 km to 0.42 at 1 x 1 km. Fractional bias also improved from -0.46 at the coarse grid
resolution to -0.39 at the finest scale. Rural predictions of $PM_{2.5}$ were also better with
increasing resolution in the summer. Fractional error decreased from 0.31 to 0.22 while
fractional bias decreased from +0.05 to -0.05.
Larger improvements are seen with increasing resolution during the summer when
compared to measurements from low-cost sensors. Starting from a large negative bias of
-5.4 µg m$^{-3}$ (fractional bias of -0.48) at the 36 x 36 km resolution, performance consistently
improved with each increasing resolution step with the bias eventually reaching -3.7 µg
m$^{-3}$ (fractional bias of -0.27) at the 1 x 1 km. There was also a reduction in fractional error
from 0.52 at the coarse to 0.41 at the fine 1 x 1 km resolution. These metrics are
encouraging, although they are likely impacted by an overprediction of the urban-rural
gradient, similar to winter. Improvement of the secondary $PM_{2.5}$ predictions is still the
largest source of error between predictions and this source of measurements.

**6 Evaluation of Novel Emissions Surrogates**

For commercial cooking, the normalized restaurant count was used to distribute the
emissions from the sector in space within the 1 x 1 km domain. Geographical information
was collected for all restaurant locations in the inner domain from the Google Places
Application Programming Interface. This includes southwestern Pennsylvania as well as



parts of eastern Ohio and northern West Virginia. To allocate on-road traffic emissions, the
output from the traffic model of Ma et al. (2020) was used. This model simulated hourly
traffic using data from the Pennsylvania Department of Transportation sites located
throughout the inner modeling domain. Changes in the spatial distribution of cooking and
on-road traffic emissions are illustrated in the supplementary material (Fig. S5-S8). These
novel emissions surrogates resulted in larger emissions of both traffic and cooking in the
downtown area. In the case of on-road traffic, major highways in the inner domain are
emphasized with the new surrogates.

For both February and July 2017, the largest observed change when using the novel

surrogates is an increase in predicted $PM_{2.5}$ of around 3 µg m$^{-3}$ in the downtown Pittsburgh
area (Fig. 7). Differences in predicted $PM_{2.5}$ concentrations outside of the urban areas of
the inner domain are very small (less than 0.5 µg m$^{-3}$ in magnitude).

Model performance at 1 x 1 km resolution is detailed in Table 4. Negligible changes

in performance were seen using EPA regulatory $PM_{2.5}$ data in February 2017. Small
improvements were seen at regulatory sites in July 2017, where fractional error was
reduced from 51% to 48% and fractional bias increased from -43% to -39%. A positive
shift in fractional bias was seen with the use of the new surrogates during both periods
when compared to low-cost sensor measurements, resulting in a modest overprediction of
$PM_{2.5}$ in the winter (+0.24 fractional bias) and a modest underprediction of $PM_{2.5}$ in the
summer (-0.27 fractional bias). The larger changes when compared to the low-cost sensor
measurements are a result of the location of the low-cost sensors in urban areas, where the
new surrogates predicted elevated $PM_{2.5}$ mass concentrations.

**7 Conclusions**

We applied PMCAMx-v2.0 over southwestern Pennsylvania during February and

July 2017 at grid resolutions of 36 km, 12 km, 4 km and 1 km. Emissions were calculated
for the relevant grids by using the spatial surrogates provided along with the 2011 NEI for
all emissions sectors except traffic and cooking, for which 1 x 1 km spatial surrogates were
developed.

PMCAMx predicts winter sulfate, elemental carbon and organic aerosol

concentrations with fractional biases below 10% at high resolution. Nitrate concentrations
are overpredicted (bias +1.4 µg m$^{-3}$) following the trend of previous studies in both the US





and Europe. Agreement with total $PM_{2.5}$ measurements is also encouraging with a
fractional bias of +5%. Variability between urban and rural predictions of local pollutants
EC and organic aerosol (OA) are reproduced well in the winter period. Underpredictions
of summer OA concentrations led to underpredictions of total $PM_{2.5}$ mass. Summer sulfate
is reproduced with fractional bias of -21% and elemental carbon (EC) is predicted with
fractional bias of -33%. Nitrate is similarly overpredicted in the summer with fractional
bias of +70% although with a much smaller magnitude than in the winter (+0.4 µg m$^{-3}$).
Improvement of the treatment of dust in the model is required to better model the
distribution of particulate nitrate between $PM_{2.5}$ and $PM_{10}$ modes. Differences between
urban and rural EC is also predicted well in the summer, while OA is predicted to vary
little between urban and rural locations. This is indicative of a greater contribution of
secondary species to OA during this period. Improvements to SOA formation chemistry
within the model, particularly from biogenic sources outside of the inner modeling domain,
will likely have a significant impact on $PM_{2.5}$ predictions around the city of Pittsburgh.
This, along with the improvement of dust treatment in the model, are topics of future work
for model improvement.

$PM_{2.5}$ prediction performance improved in almost all cases when increasing the

resolution from 36 km to 1 km. Underpredictions at urban sites and overpredictions at rural
sites were reduced at the same time. This is true when comparing against measurements
from regulatory sites as well as low-cost monitors. The improved performance here is
evidence of the enhanced ability of the model to capture important urban-rural gradients in
$PM_{2.5}$ pollution by increasing the resolution of predictions to 1 x 1 km.

*Code Availability.* The PMCAMx-v2.0 code is available in Zenodo at
https://doi.org/10.5281/zenodo.6772851 (Dinkelacker et al., 2022). License (for files):
GNU General Public License v3.0.

*Author contributions.* BTD performed the PMCAMx simulations, analyzed the results, and
wrote the manuscript. PGR wrote the code for data analysis, prepared anthropogenic
emissions and other inputs for the PMCAMx simulations, and assisted in writing the
manuscript. IK set up the WRF simulations and assisted in the preparation of the



meteorological inputs. SNP and PJA designed and coordinated the study and helped in the
writing of the paper. All authors reviewed and commented on the manuscript.

*Competing Interests.* The authors declare that they have no conflict of interest.
*Financial support.* This work was supported by the Center for Air, Climate, and Energy
Solutions (CACES) which was supported under Assistance Agreement No. R835873
awarded by the U.S. Environmental Protection Agency and the Horizon-2020 Project
REMEDIA of the European Union under grant agreement No 874753.

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





**Table 1.** Comparison of daily average high-resolution PMCAMx-v2.0 predictions with
daily EPA-CSN measurements during February and July 2017.

| February 2017 | | | | | | |
|---|---|---|---|---|---|---|
| | **Sulfate** | **Nitrate** | **Ammon.** | **Elemental Carbon** | **Organic Aerosol** | **PM$_{2.5}$** [a] |
| Measured Avg. ($\mu$g m$^{-3}$) | 1.92 | 1.51 | 0.91 | 1.08 | 4.37 | 10.34 |
| Predicted Avg. ($\mu$g m$^{-3}$) | 1.70 | 2.90 | 1.62 | 0.94 | 3.68 | 10.52 |
| Error ($\mu$g m$^{-3}$) | 0.79 | 1.54 | 1.03 | 0.78 | 2.15 | 3.02 |
| Fractional Error | 0.41 | 0.83 | 0.96 | 0.71 | 0.53 | 0.30 |
| Bias ($\mu$g m$^{-3}$) | -0.22 | 1.40 | 0.71 | -0.14 | -0.68 | 0.18 |
| Fractional Bias | -0.02 | 0.81 | 0.83 | -0.08 | -0.01 | 0.05 |


| July 2017 | | | | | | |
|---|---|---|---|---|---|---|
| | **Sulfate** | **Nitrate** | **Ammon.** | **Elemental Carbon** | **Organic Aerosol** | **PM$_{2.5}$** [a] |
| Measured Avg. ($\mu$g m$^{-3}$) | 2.04 | 0.26 | 0.53 | 0.74 | 4.46 | 11.24 |
| Predicted Avg. ($\mu$g m$^{-3}$) | 1.60 | 0.68 | 0.79 | 0.56 | 2.67 | 7.26 |
| Error ($\mu$g m$^{-3}$) | 1.12 | 0.45 | 0.39 | 0.39 | 2.46 | 4.67 |
| Fractional Error | 0.62 | 0.82 | 0.62 | 0.60 | 0.67 | 0.49 |
| Bias ($\mu$g m$^{-3}$) | -0.44 | 0.42 | 0.26 | -0.18 | -1.85 | -4.01 |
| Fractional Bias | -0.21 | 0.70 | 0.44 | -0.33 | -0.47 | -0.39 |

[a] Measurements from the regulatory EPA monitors.





**Table 2.** Comparison of daily average PMCAMx-v2.0 predicted $PM_{2.5}$ concentrations
during February and July 2017 with daily measurements from 17 EPA regulatory
monitors.

| | 36 x 36 km | 12 x 12 km | 4 x 4 km | 1 x 1 km |
|---|---|---|---|---|
| **February 2017** | | | | |
| Measured Avg. ($\mu$g m$^{-3}$) | 10.34 | 10.34 | 10.34 | 10.34 |
| Predicted Avg. ($\mu$g m$^{-3}$) | 9.78 | 9.68 | 10.49 | 10.52 |
| Error ($\mu$g m$^{-3}$) | 3.35 | 3.16 | 3.04 | 3.02 |
| Fractional Error | 0.34 | 0.32 | 0.30 | 0.30 |
| Bias ($\mu$g m$^{-3}$) | -0.56 | -0.66 | 0.15 | 0.18 |
| Fractional Bias | -0.09 | -0.10 | 0.06 | 0.05 |
| | | | | |
| **July 2017** | | | | |
| Measured Avg. ($\mu$g m$^{-3}$) | 11.24 | 11.24 | 11.24 | 11.24 |
| Predicted Avg. ($\mu$g m$^{-3}$) | 6.90 | 6.86 | 7.26 | 7.23 |
| Error ($\mu$g m$^{-3}$) | 4.89 | 5.05 | 4.67 | 4.65 |
| Fractional Error | 0.53 | 0.53 | 0.49 | 0.48 |
| Bias ($\mu$g m$^{-3}$) | -4.34 | -4.39 | -3.98 | -4.01 |
| Fractional Bias | -0.45 | -0.47 | -0.39 | -0.39 |






**Table 3.** Comparison of daily average PMCAMx-v2.0 predicted PM$_{2.5}$ concentrations
during February and July 2017 with daily low-cost sensor (RAMP) measurements.

|  | 36 x 36 km | 12 x 12 km | 4 x 4 km | 1 x 1 km |
|---|---|---|---|---|
| **February 2017** | | | | |
| Measured Avg. ($\mu$g m$^{-3}$) | 11.65 | 11.65 | 11.65 | 11.65 |
| Predicted Avg. ($\mu$g m$^{-3}$) | 10.23 | 11.64 | 12.04 | 13.50 |
| Error ($\mu$g m$^{-3}$) | 4.53 | 4.53 | 4.51 | 5.12 |
| Fractional Error | 0.33 | 0.33 | 0.34 | 0.37 |
| Bias ($\mu$g m$^{-3}$) | -1.43 | -0.02 | 0.4 | 1.85 |
| Fractional Bias | -0.02 | <0.01 | 0.14 | 0.24 |
| | | | | |
| **July 2017** | | | | |
| Measured Avg. ($\mu$g m$^{-3}$) | 12.59 | 12.59 | 12.59 | 12.59 |
| Predicted Avg. ($\mu$g m$^{-3}$) | 7.19 | 7.44 | 8.06 | 8.83 |
| Error ($\mu$g m$^{-3}$) | 5.60 | 5.70 | 5.29 | 4.89 |
| Fractional Error | 0.51 | 0.51 | 0.46 | 0.42 |
| Bias ($\mu$g m$^{-3}$) | -5.40 | -5.15 | -4.53 | -3.76 |
| Fractional Bias | -0.48 | -0.43 | -0.36 | -0.27 |






**Table 4.** Performance of daily average predicted total PM$_{2.5}$ concentrations compared to daily measurements from regulatory sites and low-cost sensors with the use of old surrogates and new surrogates for on-road traffic and commercial cooking.

| | February 2017 | | | |
|---|---|---|---|---|
| | **Old Surrogates** | | **New Surrogates** | |
| | Regulatory network | Low-cost sensors | Regulatory network | Low-cost sensors |
| **Observed Average (µg m$^{-3}$)** | 10.34 | 11.65 | 10.34 | 11.65 |
| **Predicted Average (µg m$^{-3}$)** | 10.23 | 11.32 | 10.52 | 13.50 |
| **Error (µg m$^{-3}$)** | 2.94 | 4.12 | 3.02 | 5.12 |
| **Fractional Error** | 0.29 | 0.31 | 0.30 | 0.37 |
| **Bias (µg m$^{-3}$)** | -0.11 | -0.33 | 0.18 | 1.85 |
| **Fractional Bias** | -0.04 | 0.08 | 0.05 | 0.24 |

| | July 2017 | | | |
|---|---|---|---|---|
| | **Old Surrogates** | | **New Surrogates** | |
| | Regulatory network | Low-cost sensors | Regulatory network | Low-cost sensors |
| **Observed Average (µg m$^{-3}$)** | 11.24 | 12.58 | 11.24 | 12.58 |
| **Predicted Average (µg m$^{-3}$)** | 7.09 | 7.98 | 7.26 | 8.83 |
| **Error (µg m$^{-3}$)** | 4.91 | 5.32 | 4.67 | 4.89 |
| **Fractional Error** | 0.51 | 0.47 | 0.49 | 0.42 |
| **Bias (µg m$^{-3}$)** | -4.33 | -4.61 | -4.01 | -3.76 |
| **Fractional Bias** | -0.43 | -0.37 | -0.39 | -0.27 |






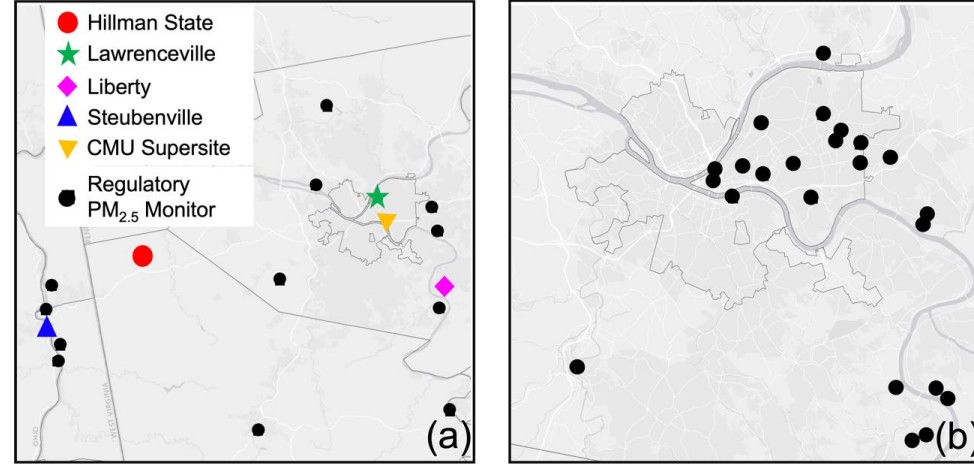

**Figure 1.** Monitoring sites. (**a**) Particulate matter speciation measurement sites from EPA-
CSN and PM$_{2.5}$ regulatory monitors. (**b**) low-cost sensor sites.






**Figure 2.** Comparison of daily average PMCAMx-v2.0 predicted concentrations of PM$_{2.5}$
(a) sulfate, (b) nitrate, (c) ammonium, (d) elemental carbon, and (e) organic aerosol with
daily measurements from EPA-CSN sites during February 2017.



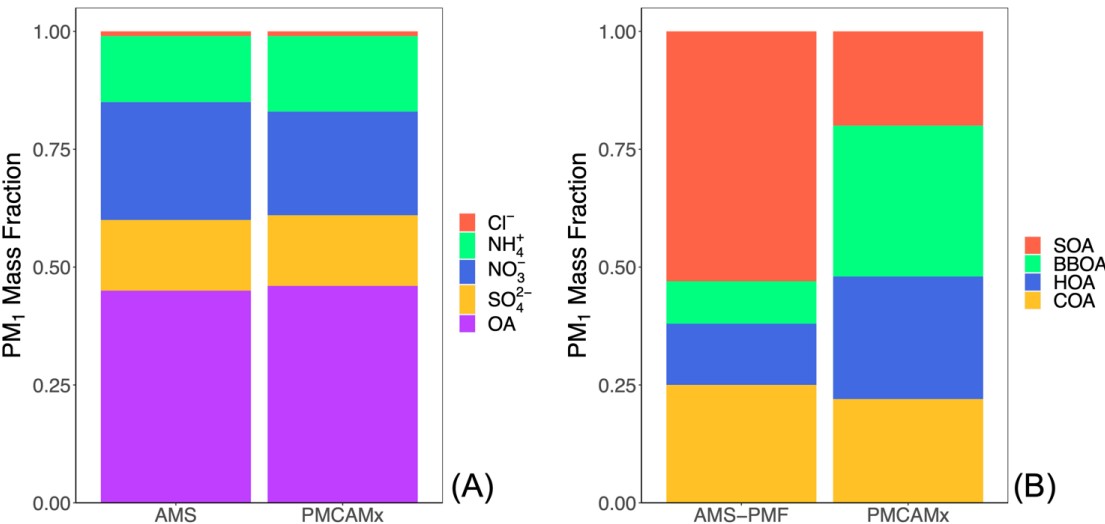

**Figure 3.** (**a**) Comparison of PMCAMx-v2.0 predicted composition of $PM_1$ with the corresponding AMS measurements at the CMU site and (**b**) organic aerosol composition based on the PMF analysis of the AMS measurements and predicted composition.


**Figure 4.** Comparison of PMCAMx-v2.0 predicted concentrations of PM$_{2.5}$ (a) sulfate, (b)
nitrate, (c) ammonium, (d) elemental carbon, and (e) organic aerosol with measurements
from EPA-CSN sites during July 2017.




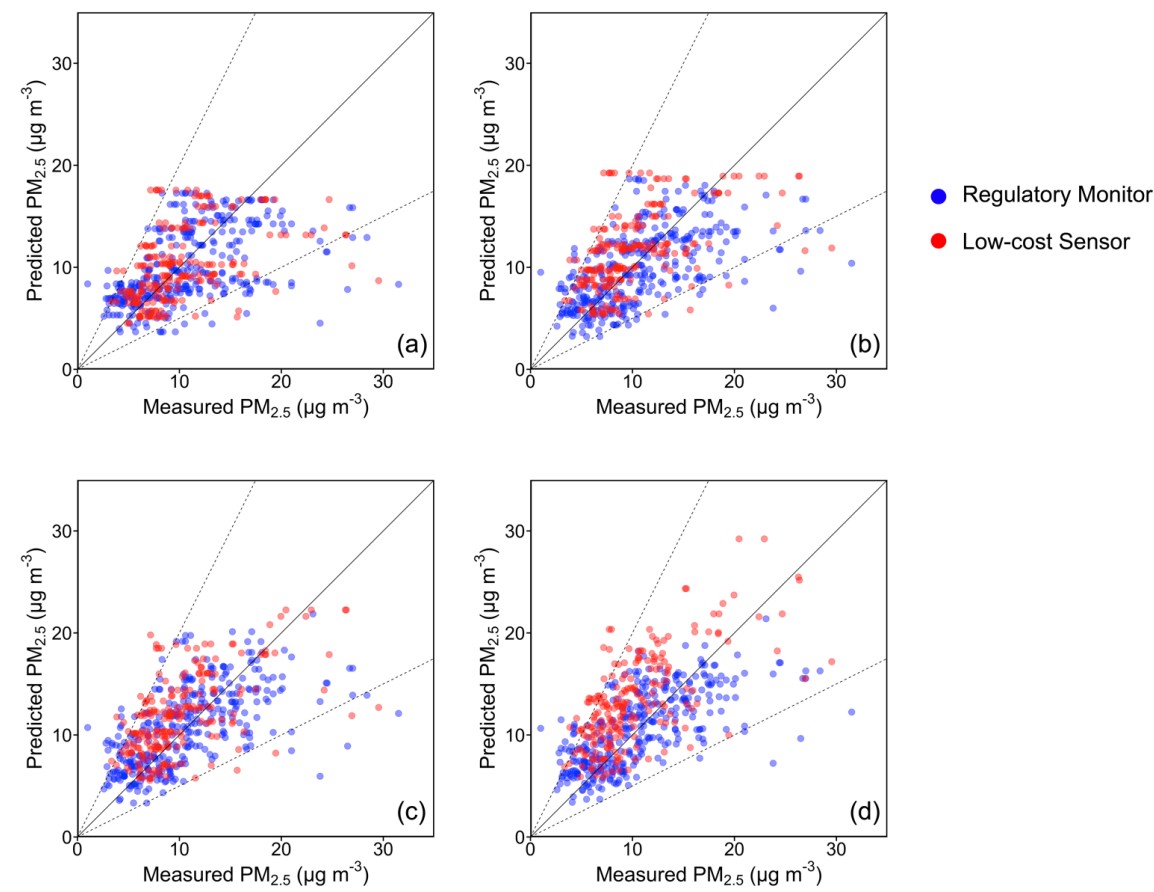

**Figure 5.** Comparison of daily average PMCAMx-v2.0 predicted concentrations of $PM_{2.5}$
with daily regulatory measurements and daily low-cost sensor measurements at (a) 36 x
36, (b) 12 x 12, (c) 4 x 4, and (d) 1 x 1 km during February 2017.


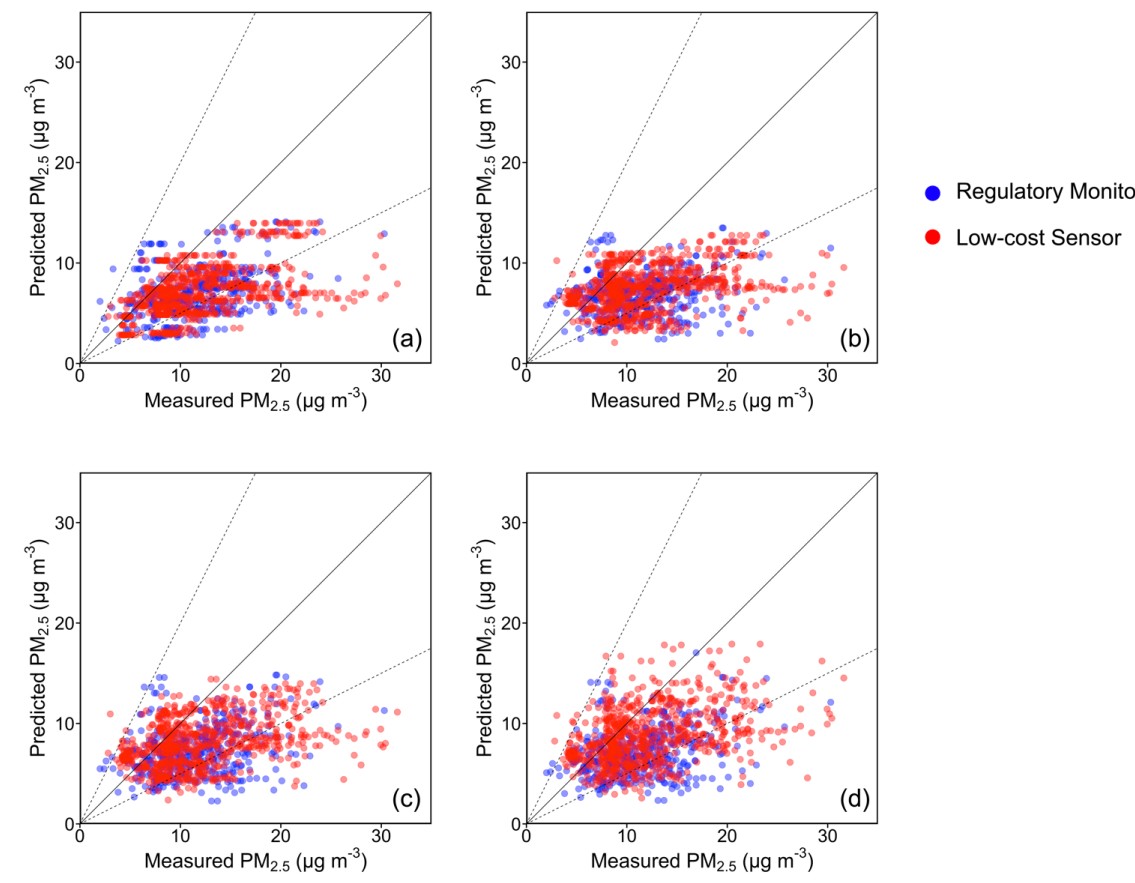

**Figure 6.** Comparison of daily average PMCAMx-v2.0 predicted concentrations of PM$_{2.5}$
with daily regulatory measurements and daily low-cost sensor measurements at (a) 36 x
36, (b) 12 x 12, (c) 4 x 4, and (d) 1 x 1 km during July 2017.

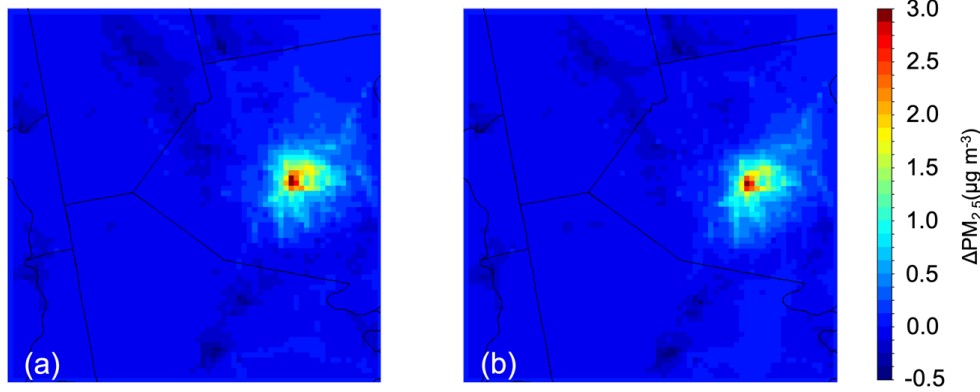

**Figure 7.** Difference between predicted monthly average PM$_{2.5}$ mass concentration when using novel surrogates and original surrogates in (**a**) February 2017 and (**b**) July 2017 for the 1 x 1 km resolution simulation grid. A positive value indicates a higher concentration predicted with the novel surrogates.