# Peer review of "Evaluation of high-resolution predictions of fine particulate matter and its composition in an urban area using PMCAMx-v2.0"

_Geoscientific Model Development, 2022_

## Author Response (AR1)

**Responses to the Comments of the Referees**

**Referee #1**

**Major comments**

**(1)** The analysis presented and discussed is based on a modelling run for only two months – February and July – and the discussion and conclusions are drawn for Winter and Summer periods. An example is the sentence in line 327 "Total PM$_{2.5}$ mass concentrations are underpredicted in the summer period". It is clear that those months were chosen due to the availability of measurements, particularly PM$_{2.5}$ compounds. Notwithstanding, the study would be much more robust and conclusion better supported if the modelling period was at least one year. Results could be assessed with the observations from fixed monitoring sites for the whole year, with a focus on those two months for which more data is available.

We do agree that longer evaluations are always better. However, significant resources are required to develop the emission inventory used for these PMCAM$_x$ simulations, so two months in two different seasons were chosen as a compromise. These simulations can provide sufficient information, so that the ability of the model to capture PM$_{2.5}$ differences due to the seasonal variability in major emissions sources and meteorology can be adequately assessed. The primary goal of this study is to evaluate the predictions of PM$_{2.5}$ concentrations at high spatial resolution (1 km x 1 km), produced using a corresponding high spatial resolution source-resolved emissions inventory. The high-resolution, source-resolved, and species resolved PM$_{2.5}$ predictions evaluated in this study are novel and require significant evaluation against available measurements which can be sparse (especially for PM$_{2.5}$ component analysis). Another factor in our decision has been the connection of the present work with the corresponding analysis and conclusions of Garcia Rivera et al. (2022) study who also studied the corresponding periods. A brief explanation and discussion of this point has been added in the revised paper.

**(2)** The authors refer that "Garcia Rivera et al. (2022) investigated the effects of increasing grid resolution of model inputs and CTM output on source resolved predictions of PM$_{2.5}$ concentration and population exposure at 36 km, 12 km, 4 km, and 1 km". Is the current work based on the simulations performed by Garcia Rivera et al. (2022) or are there additional runs in the present work? Are the runs referred in lines 309 – 310 examples of additional simulations? What exactly is the novelty of this study compared to (Garcia Rivera et al.,2022)? This should be made clear in the manuscript and perhaps the "Model Application" section could be shortened since it is already explained in the published paper.

This study uses the results of some of the simulations of Garcia Rivera et al. (2022) and adds several additional simulations. For example, additional simulations were performed to quantify the impact of the proposed emissions surrogates for traffic and cooking. The work of Garcia Rivera et al. (2022) is an analysis of the impact of increasing grid resolution on predicted human exposure to PM$_{2.5}$ and the corresponding links with the PM$_{2.5}$ sources. Garcia Rivera et al. (2022) did not address model performance and the corresponding challenges related to the different types of the available measurements. This evaluation is the primary purpose of the present study. The Model Application section now contains only the most essential information about the simulations so that the interested reader does not need to consult Garcia Rivera et al. (2022) to understand the major features of our simulations. These points have been clarified in the revised manuscript text.

**Specific comments**

**(3)** The abstract is too long and has many details of the analysis of results that are unnecessary in an abstract. Conversely, the Conclusions lack a more comprehensive and focused text on the added value and limitations of this work and concrete ideas for future work.

The content of the abstract and conclusions have been reassessed and changes have been made in the manuscript. Additional clarity on the scope of the work has also been highlighted in the revised manuscript.

**(4)** Section 3 has a subsection 3.1, that would only make sense if a 3.2 exists. Please restructure this section or remove the subsection title. One idea is to join sections 2 and 3 into one section.

This has been fixed in the manuscript.

**(5)** Figure 1 – Please include the metric scale and draw the inner domain to allow to have an idea of where stations are located in relation to the modelling domains.

Figure 1 (a) shows fully the entire inner domain. Figure 1 (b) shows the City of Pittsburgh limits, which are also shown in Figure 1 (a) to provide the reader with a scale reference. These notes have been included in the revised caption for clarity.

**Technical Corrections**

**(6)** Line 323 - The reference (Kodros et al., 2021) is missing in the reference list.

The missing reference has been added.

**(7)** References Day et al., 2015, Skyllakou et al., 2021, included in the reference list are not cited in the text.

These unused references have been removed.

**Referee #2**

**Comments**

**(1)** While it is clear that the different spatial distribution has an impact on the distribution of emissions for commercial cooking and on-road traffic, how are these two sectors typically distributed in the default surrogates? Please consider elaborating in Section 3 to provide some context to the reader.
The spatial distributions of cooking and traffic emissions using both the new and old surrogates are illustrated in the Figures S5, S6, S7 and S8 located within the supplementary material. This was not made clear in the main text and some text has been added to both explain the approach used and to also point the reader to these additional figures.

**(2)** In the PMCAMx model simulations, are there removal processes such as dry deposition and wet scavenging? If so, how were these handled between the different grid resolutions? Are these something that must be interpolated from WRF or is it something that PMCAMx explicitly captures?
These processes are included in PMCAMx, and this has been clarified in the included model description. Wet deposition is dependent on precipitation, which indeed relies upon the meteorological input data. The need to interpolate meteorology is a limitation of the study, however the evaluation process described in the supplementary material (Figures S1 and S2 with corresponding discussion) provides confidence in this specific application.

**(3)** The WRF simulations were conducted at 12 km resolution with the information interpolated to higher resolutions of 1 km and 4 km. First, what was the procedure to create 36 km inputs for PMCAMx? Second, why was this design choice made rather than simulating WRF at 36 km and interpolating for all higher resolution grids? Or perhaps simulate meteorology at 4 km with interpolating to 1 km grid and extrapolating to 12 and 36 km grids?
The WRF simulations and PMCAMx simulations use the same domain, meaning we can directly aggregate the nine 12 km x 12 km cells that make up a single 36 km x 36 km cell to create input data for the coarse grid. For the higher resolution grids, we use the interpolation process used in previous work (Fountoukis et al., 2013) that has been evaluated extensively for use in higher resolution air quality studies. An evaluation for the specific meteorological data used in this study is provided for the reader in the supplementary material (Figures S1 and S2 with corresponding discussion).

**(4)** For those unfamiliar with emissions processing and emission surrogates, it may prove useful to further emphasis how using different surrogates merely change the spatial distribution of the total amount. The authors perhaps made this clear in Section 6 and the Supplementary material, but the reader could benefit from this stated more clearly and earlier.
A short description of this is now provided in Section 3 (Model Application). A clarifying discussion about how these surrogates might change the emissions provided to the model has been added.

**(5)** Related to the previous point, at times in the paper it reads like the novel surrogates for commercial cooking and on-road emissions were applied to all grid resolution configurations and other times it reads like they were only applied to 1 and 4 km simulations (see Line 183-184). They were used for the 1 km simulations. Conflicting statements have been removed and a clarifying note has been added to the caption of Table 4 highlighting that these results are for the high-resolution simulations only.

**(6)** There is no mention of the novel emission surrogates for commercial cooking and on-road emissions in the abstract. I feel like this point could use some emphasis in terms of mentioning that these were used and developed and briefly mention the evaluation of these novel surrogates. A sentence has been added to the abstract, noting the use and evaluation of new surrogates for these sources at high spatial resolution.

**(7)** The paper could in general benefit from making it more clear how this study is both related and also different from Rivera et al 2022. These studies are of course related. This work is meant to be a detailed quantitative evaluation of the simulation results used to investigate the changes in predicted concentration and exposure with increasing resolution in Garcia Rivera et al. (2022). None of this evaluation is included in the previous work. A clarifying statement has also been added, noting the additional simulations performed to quantify any differences in the use of the novel spatial surrogates for commercial cooking and on-road traffic.

**Minor comments**

**(8)** Table 4 could benefit from explicitly mentioning in the caption that it is for the $1 \times 1$ km resolution. The recommended change has been made in Table 4.

**(9)** In supplementary material, the captions for Figure S7 and S8 should be checked and corrected to avoid any confusion. Figures S7 and S8 captions incorrectly start off with mentioned commercial cooking and also both for February 2017. Presumably, S7 and S8 are for on-road traffic for February and July respectively as they mention using simulated traffic approach. These are correctly identified typos. They have been corrected in the manuscript.

---

## Author Response (AR2)

**Responses to the Editor's Comments**

**(1)** Please clean the source code archive (PMCAMx_v2.0.tar.gz) from any temporary or compiled binary files, which include: 325 *.o files and several *.swp/*.swo files (.trap.f.swp, CMC/.rateslo5.f.swo, CMC/.rxnrate5.f.swp, IO_bin/.readar.f.swp).
These files have been removed from the archive and re-uploaded to the zenodo site.

**(2)** Should Carter 1999 be changed to Carter 2000 with additional bibliographic info (as cited, e.g., here: https://intra.engr.ucr.edu/~carter/pubs/,
https://gmd.copernicus.org/articles/10/2397/2017/gmd-10-2397-2017.pdf): Final Report to California Air Resources Board Contract 92-329 and Contract 95-308 ... 2000.
 Assuming CiteSeerX repositories are permanent, it seems that an URL can be provided for the above (please check and confirm):
https://citeseerx.ist.psu.edu/doc_view/pid/71f7566ae19ae551b3d214c0fc54250ebc399743
The Carter reference for the SAPRC mechanism has been updated as suggested.

**(3)** Please double check the year of CAMx 4.20 documentation reference; it is cited as ENVIRON 2005 elsewhere, e.g.,
on page 297 here: https://books.google.com/books?id=5rISBwAAQBAJ
This reference has been updated as suggested.

**(4)** Please double check the year of Eyth & Vukovich entry, if it shouldn't be 2015 as here:
https://www.epa.gov/sites/default/files/2015-
10/documents/2011v6_2_2017_2025_emismod_tsd_aug2015.pdf
It should be 2015. This has been updated.

**(5)** Order of references Zakoura & Pandis 2019 and 2018 should likely be switched in the reference list
This has been fixed in the manuscript.

**(6)** Please match case across ENVIRON (in the reference list) and Environ (when citing)
This has been fixed in the manuscript.

**(7)** Please add DOI for Pan et al. 2017: https://doi.org/10.1016/j.atmosenv.2017.06.026
This has been added to the manuscript.

**(8)** Please update bibliographic entry for Tsimpidi et al. 2010 so it refers to final published paper and not pre-review paper
(10.5194/acpd-9-13693-2009 -> https://doi.org/10.5194/acp-10-525-2010)
This has been fixed in the manuscript.

**(9)** The link in U.S. EPA 2002 entry no longer works
A new, valid link has been added to the reference.

**(10)** Typos:
- p4/l116: regards -> regard

- p5/l158: norther -> northern
- p12/l340: seen the -> seen in the
- p13/l382: regards -> regard
- p16/l484: need additional -> need for additional
- p18/l546: applicaiton -> application

These typos have all been corrected in the manuscript.

**(11)** Please add "USA" to all US institutional addresses in the affiliations

This has been added to the author list.

**(12)** Please aim at making the Supplement figures essentially readable without interpreting colors by using different types of lines (Figs S1, S2) and symbols (Figs 5, 6, S3, S4)

Different lines and symbols have been added to Figs S1, S2, 5, and 6. The individual datasets in Figures S3 and S4 have their own axes for interpretability.